# Synthesis and Structures of Lead(II) Complexes with Hydroxy-Substituted *Closo*-Decaborate Anions

Evgenii Yu. Matveev [1,2], Varvara V. Avdeeva [2,*], Alexey S. Kubasov [2], Konstantin Yu. Zhizhin [2], Elena A. Malinina [2] and Nikolay T. Kuznetsov [2]

[1] Institute of Fine Chemical Technologies Named after M. V. Lomonosov, MIREA—Russian Technological University, Vernadskogo pr. 86, Moscow 119571, Russia

[2] Kurnakov Institute of General and Inorganic Chemistry, Russian Academy of Sciences, Leninskii pr. 31, Moscow 119991, Russia

* Correspondence: avdeeva.varvara@mail.ru

**Abstract:** Mixed-ligand lead(II) complexes with 2,2′-bipyridyl and $[B_{10}H_9OH]^{2-}$ or monosubstituted hydroxy-substituted *closo*-decaborate anions with a pendant hydroxy group, separated from the boron cage by an alkoxylic spacer of different lengths $[B_{10}H_9O(CH_2)_xO(CH_2)_2OH]]^{2-}$ ($x = 2$ or 5) have been synthesized. Compounds have been characterized by IR and multinuclear NMR spectroscopies. The structures of binuclear complex $[Pb(bipy)_2[B_{10}H_9OH]]_2 \cdot CH_3CN$ (**1**·CH_3CN), mononuclear complex $[Pb(bipy)_2[B_{10}H_9O(CH_2)_2O(CH_2)_2OH]] \cdot 0.5bipy \cdot CH_3CN$ (**2**·0.5bipy·CH_3CN), and polymeric complex $[Pb(bipy)[B_{10}H_9O(CH_2)_5O(CH_2)_2OH]]_n$ (**3**) have been determined by single-crystal X-ray diffraction. In all three compounds, the co-ordination polyhedra of lead(II) are formed by N atoms from two bipy molecules, O atoms of the substituent attached to the boron cage, and BH groups of the boron cage.

**Keywords:** boron cluster anions; decahydro-*closo*-decaborate anion; decahydrido-*closo*-decaborate anion; complexes; lead; non-covalent interactions

## 1. Introduction

Boron cluster anions $[B_nH_n]^{2-}$ ($n = 10, 12$) [1–5] are a unique example of homonuclear nonmetallic clusters, which are characterized by increased stability and allow one to replace terminal hydrogen atoms by various functions [6–13]. These aromatic inorganic 3D cages [14–17] may be of interest as platforms for further modification; for example, *closo*-hydridoborates with exopolyhedral oxonium substituents can react with nucleophilic reagents to give compounds with pendant functional groups [18–27]. Moreover, boron clusters $[B_nH_n]^{2-}$ containing a thiol substituent can participate in alkylation and acylation reactions at the sulfur atom [28–30].

Derivatives of *closo*-hydridoborates, in addition to their classical applications in ${}^{10}B$ neutron capture therapy [31–37], the preparation of electrolytes [38], MOFs [39,40] etc., have recently attracted considerable interest in the synthesis of complex compounds. They usually act as counterions in metal complexation [41–46], but their ability to play the role of polydentate and polytopic ligands is of growing interest [47–49]. As generalized in recent reviews [50,51], substituted *closo*-borates have extensive complexation capacity due to the presence of various types of co-ordination centers: BH groups of the boron cluster as a soft Pearson base and an introduced functional group which can contain functions of different natures (hard and soft bases). Varying the structure of the exopolyhedral fragment, including its Pearson hardness, allows one to design complexes with desired structures and properties. The majority of complexes containing co-ordinated *closo*-borate anions are formed by metals acting as Pearson's soft acids (first of all, copper(I), silver(I), and lead(II) (see reviews [50,51] and references thereof)). The presence of a spacer containing

donor atoms and linking the boron cage and a pendant functional group makes it possible to expand the co-ordination ability of substituted *closo*-hydridoborates. Substituted derivatives of the boron cluster anions can form complexes with boron cages co-ordinated via BH-groups to form 3c2e MHB bonds, via functions of the substituent, or a combined co-ordination mode can be expected with both BH groups and the substituent to be co-ordinated. These studies are of both fundamental and practical interest, for example, for the synthesis of functional boron-containing materials [52–55].

Few lead(II) complexes with substituted derivatives of the boron clusters $[B_nH_n]^{2-}$ ($n$ = 10, 12) with the exopolyhedral B–O bonds have been reported; compounds have been isolated in the presence of bipy. The structures of lead(II) complexes $[Pb(DMF)L[2-B_{10}H_9OH]]\cdot DMF$ (CCDC 182269) [56], $[PbL[2-B_{10}H_9O(CH_2CH_2)_2OEt]]2\cdot0.5DMF$ (CCDC 282062) [57], $(Ph_4P)[PbL[2-B_{10}H_9OC(O)CH_3)_2]_2$(CCDC 204827) [58], and $[PbL_2[2,8-B_{10}H_8(OC(O)CH_3)_2)]]$ (CCDC 204828) [56] (L = bipy) were determined by X-ray diffraction. Complexes were isolated when salts of the corresponding *closo*-decaborate derivatives reacted with solid $Pb(NO_3)_2$ in the presence of bipy; DMF solvates were isolated when recrystallizing the obtained complexes. In all compounds under discussion, lead(II) co-ordinates the boron cages via 3c2e PbHB bonds and/or Pb–O bonds with the substituent introduced into the boron cluster.

Here, we synthesized lead(II) complexes with hydroxy-substituted derivatives $[2-B_{10}H_9OH]^{2-}$ and $[B_{10}H_9ROH]^{2-}$ (where spacer R = $O(CH_2)_xO(CH_2)_2$, $x$ = 2 or 5) in the presence of the organic ligand bipy; their structures were determined by X-ray diffraction and interactions formed between elements of the structures were discussed.

## 2. Results and Discussion

The monohydroxy *closo*-decaborate anion $[2-B_{10}H_9OH]^{2-}$ was obtained starting from the $[B_{10}H_{10}]^{2-}$ anion; derivatives with the pendant hydroxy group of the composition $[2-B_{10}H_9O(CH_2)_2O(CH_2)_2OH]^{2-}$ and $[2-B_{10}H_9O(CH_2)_5O(CH_2)_2OH]^{2-}$ were obtained by opening cyclic substituents in monoanionic derivatives $[2-B_{10}H_9O(CH_2)_4O]^-$ and $[2-B_{10}H_9O(CH_2)_5]^-$ respectively (Scheme 1).

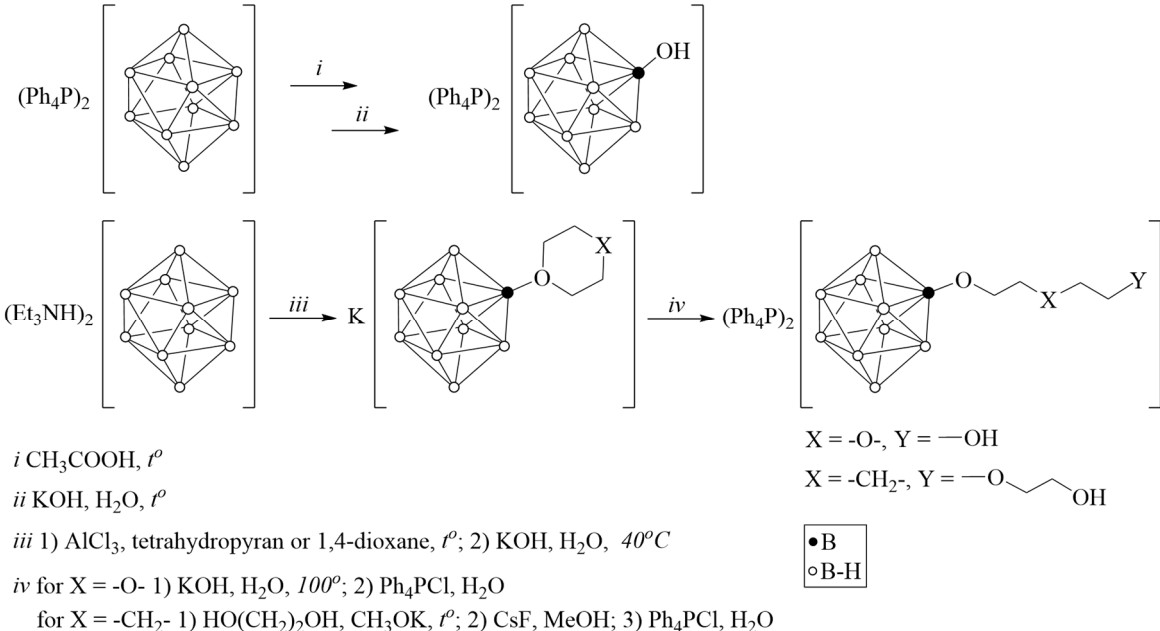

$i$ CH$_3$COOH, $t^o$

$ii$ KOH, H$_2$O, $t^o$

$iii$ 1) AlCl$_3$, tetrahydropyran or 1,4-dioxane, $t^o$; 2) KOH, H$_2$O, $40^oC$

$iv$ for X = -O- 1) KOH, H$_2$O, $100^o$; 2) Ph$_4$PCl, H$_2$O

for X = -CH$_2$- 1) HO(CH$_2$)$_2$OH, CH$_3$OK, $t^o$; 2) CsF, MeOH; 3) Ph$_4$PCl, H$_2$O

**Scheme 1.** Synthesis of substituted derivatives $[2-B_{10}H_9OR]^{2-}$.

The formation of the *closo*-decaborate anion derivative with a pendant functional group $[B_{10}H_9O(CH_2)_5O(CH_2)_2OH]^{2-}$ was monitored using $^{11}B$ NMR spectroscopy. At the first stage, a tetrahydropyran molecule was attached to the *closo*-decaborate anion as an exopolyhedral substituent to form a monoanion derivative. The $^{11}B$ {$^1H$} NMR spectrum of the

obtained compound (Figure S1) contains a signal at 6.3 ppm corresponding to the *ipso*-boron atom (B(2)), two signals at −0.1 and −5.9 ppm assigned to nonequivalent apical vertices (B(1) and B(10) respectively), as well as signals from other equatorial boron atoms: adjacent to the *ipso*-position (−22.0 and −23.8)) and opposite one (−29.9 and −31.1 ppm from B(7,8) and B(4) atoms, respectively). At the second stage, the $[B_{10}H_9O(CH_2)_5]^-$ anion reacted with the nucleophilic reagent $HO(CH_2)_2OK$ to form the $[B_{10}H_9O(CH_2)_5O(CH_2)_2OH]^{2-}$ derivative containing the pendant hydroxy group. The $^{11}B$ {$^1H$} NMR spectrum of the dianion derivative (Figure S3) remains typical for monosubstituted derivatives of the *closo*-decaborate anion with characteristic changes indicating the opening of the oxonium cyclic substituent. In particular, in the spectrum of the $[B_{10}H_9O(CH_2)_5O(CH_2)_2OH]^{2-}$ derivative, there is a shift of the signal from the *ipso*-boron atom towards a strong field up to −1.3 ppm, convergence of signals from nonequivalent apical peaks (−3.1 and −3.9 ppm), as well as rearrangement of signals from other equatorial boron atoms. Such changes unambiguously indicate the formation of 2-substituted derivatives of the $[B_{10}H_{10}]^{2-}$ anion with a pendant functional group, but cannot be used to determine its structure. This can be achieved by analyzing the $^1H$ NMR spectra of the resulting compounds (Figures S2 and S4). For example, the $^1H$ NMR spectrum of the $[B_{10}H_9O(CH_2)_5O(CH_2)_2OH]^{2-}$ anion contains seven triplets from five methylene groups of the alkoxy spacer chain and two methylene groups of the–$O(CH_2)_2OH$ fragment attached (Figure S4).

Substituted derivatives with the OH group were further used in lead(II) complexation in the presence of bipy. Complexation reactions were carried out according to Scheme 2.

**Scheme 2.** Synthesis of lead(II) complexes **1–3** with substituted derivatives $[2-B_{10}H_9R]^{2-}$.

Solid lead(II) nitrate was added to a solution of salt $(Ph_4P)_2[B_{10}H_9R]$ in acetonitrile (R = OH, $O(CH_2)_2O(CH_2)_2OH$, or $O(CH_2)_5O(CH_2)_2OH$). After 3 h of mixing, the unreacted $Pb(NO_3)_2$ was filtered. A lead(II) complex of bright yellow color forms in the reaction solution; we can imagine that its composition is $[Pb(CH_3CN)_x[An]]$, where [An] is one of three substituted *closo*-decaborate anions under discussion. We failed in isolating this compound because after evaporation of acetonitrile, a yellow oil is formed which does not allow one to separate reagents and products. Thus, organic ligand bipy was added to the reaction solution to saturate the co-ordination sphere of lead(II) and stabilize the lead(II) complexes formed. We succeeded in isolating single crystals of key products **1**·$CH_3CN$, **2**·0.5bipy·$CH_3CN$, and **3** which were suitable for further X-ray diffraction.

IR spectra of the obtained compounds **1–3** contain a broad intense band of stretching vibrations ν(BH) near 2460 $cm^{-1}$ assigned to unco-ordinated BH bonds and a band ν(BH)$_{MHB}$ which is observed as a shoulder in the region of 2300–2200 $cm^{-1}$. The presence of bands in the region 2300–2200 $cm^{-1}$ indicates the co-ordination of the boron cage via the MHB bond similar to those first reported for the copper(I) complex $[Cu_2[B_{10}H_{10}]]$ [59,60]. These type of interactions are widely found in complexes containing metals acting as soft bases according to Pearson's concept, such as copper(I), silver(I), and lead(II) and the boron cluster anions $[B_nH_n]^{2-}$ (*n* = 6, 10, 12) or their substitution derivatives and have been discussed in detail recently (see [50,51] and references thereof). In addition, the IR spectra of compounds **1–3** contain bands in the region 1700–600 $cm^{-1}$ which indicate the presence of co-ordinated bipy.

The structures of lead complexes **1**·CH$_3$CN, **2**·0.5bipy·CH$_3$CN, and **3** were determined by X-ray diffraction. The data obtained by IR spectroscopy, $^{11}$B, $^1$H, and $^{13}$C NMR spectroscopies (Figures S5–S7) as well as X-ray diffraction, correlate and indicate that the derivatives of the B$_{10}$ polyhedron are co-ordinated by the metal without changing the structure of the exopolyhedral substituent.

The crystallographically independent part of the monoclinic unit cell ($P2_1/c$) of compound **1**·CH$_3$CN contains complex [Pb(bipy)$_2$[B$_{10}$H$_9$OH]] and one solvate acetonitrile molecule (Figure 1a). The [B$_{10}$H$_9$OH]$^{2-}$ anion acts as a bridging ligand, thus forming binuclear complex [(bipy)$_2$Pb[μ-B$_{10}$H$_9$OH]]$_2$. The boron cluster anion is co-ordinated via the B3B4B8 faces, the B1B2 edge, and through the O atom of the hydroxy group. On the Hirschfeld surface (Figure 1b), the Pb–B contacts are marked as extensive red spots and dotted red lines. The co-ordination environment of lead thus includes four N atoms from two acetonitrile molecules, the B1B2OH edge, and the B4$^1$B5$^1$ ($^1$ 1 − x, 1 − y, 1 − z) edge of the second anion. However, it is not possible to draw a clear boundary between the face and edge co-ordination of the anion in this case (Table 1).

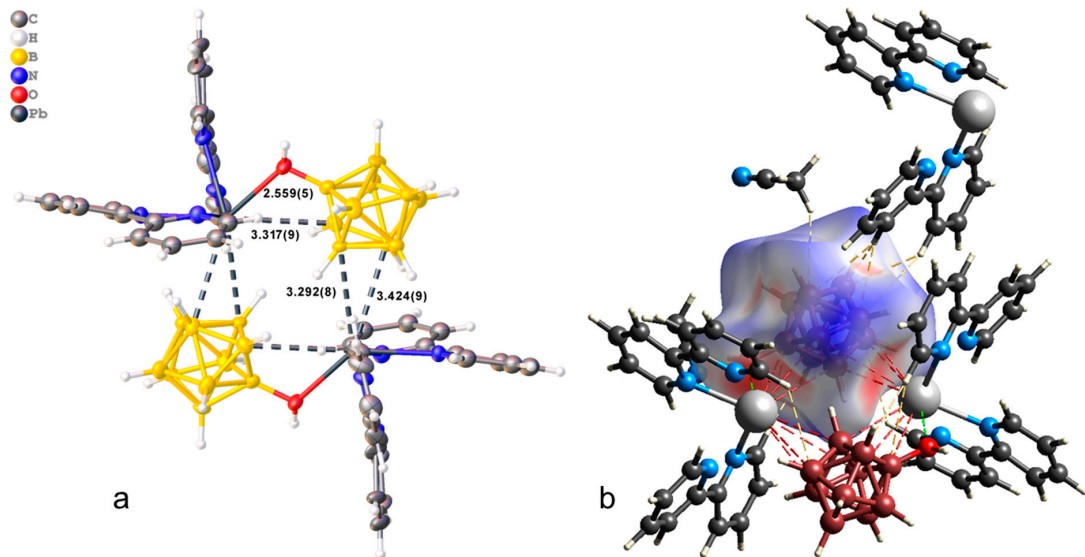

**Figure 1.** (**a**) Structure of binuclear complex [(bipy)$_2$Pb[μ-B$_{10}$H$_9$OH]]$_2$ (**a,b**) Hirschfeld surface of the [B$_{10}$H$_9$OH]$^{2-}$ anion in complex **1**·CH$_3$CN.

**Table 1.** Metal–heteroatom distances in complex **1**·CH$_3$CN.

| Bond | Distance, Å |
|:---:|:---:|
| Pb1–O1 | 2.559(5) |
| Pb1–N1 | 2.539(6) |
| Pb1–N2 | 2.622(6) |
| Pb1–N3 | 2.584(6) |
| Pb1–N4 | 2.533(6) |
| Pb1–B1 | 3.317(9) |
| Pb1–B5 | 3.721(9) |
| Pb1–B4 $^1$ | 3.424(9) |
| Pb1–B5 $^1$ | 3.424(9) |
| Pb1–B8 $^1$ | 3.740(7) |

$^1$ 1 − x, 1 − y, 1 − z.

In the packing of complex **1**·CH$_3$CN (Figure 2), binuclear complexes [(bipy)$_2$Pb[μ-B$_{10}$H$_9$OH]]$_2$ are arranged in columns parallel to axis *a* and are connected by π–π stacking interactions (centroid–centroid distance 3.800 Å, angle 10.0°, shift distance 1.931 Å). On the Hirschfeld surface of the anion (Figure 1a), it is clear that the columns are coupled to each other due to weak CH . . . HB interactions. This type of contact accounts for 79.6% of the total surface of the [B$_{10}$H$_9$OH]$^{2-}$ anion, while the H . . . Pb contacts account for 4.4% of the surface, and the Pb . . . B contacts account for 0.6% of the anion surface. The free acetonitrile molecule is located in the voids formed by four "columns" and is hydrogen-bonded to the hydroxy group (O . . . N distance 2.96 Å, N . . . H–O angle 159.1°).

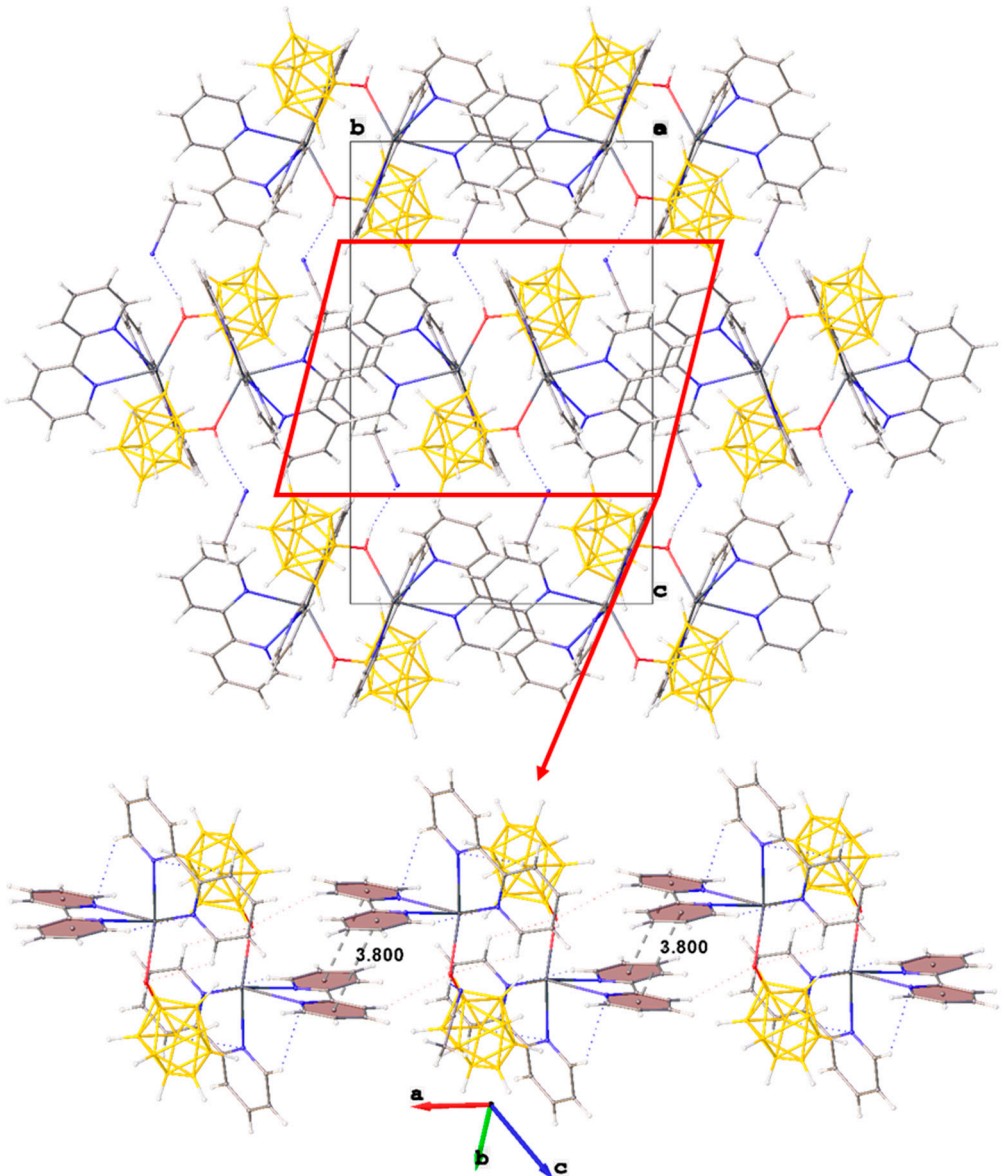

**Figure 2.** Packing diagram of **1**·CH$_3$CN and π–π stacking interactions in the crystal.

The crystallographically independent part of the triclinic unit cell of **2**·0.5bipy·CH$_3$CN (Figure 3a) contains the complex molecule [Pb(bipy)$_2$[B$_{10}$H$_9$O(CH$_2$)$_2$O(CH$_2$)$_2$OH]], half of the bipyridyl clathrate molecule, and one acetonitrile molecule. The coordination environment of lead includes four N atoms from two bipyridyl molecules, three O atoms from the exopolyhedral substituent of the [B$_{10}$H$_9$O(CH$_2$)$_2$O(CH$_2$)$_2$OH]$^{2-}$ anion, and the B6B9 edge. The Pb–X distances are given in Table 2. On the Hirschfeld surface of the boron

anion (Figure 3b), the Pb–B contacts are shown as dashed green lines, and the Pb–P contacts as dashed red lines.

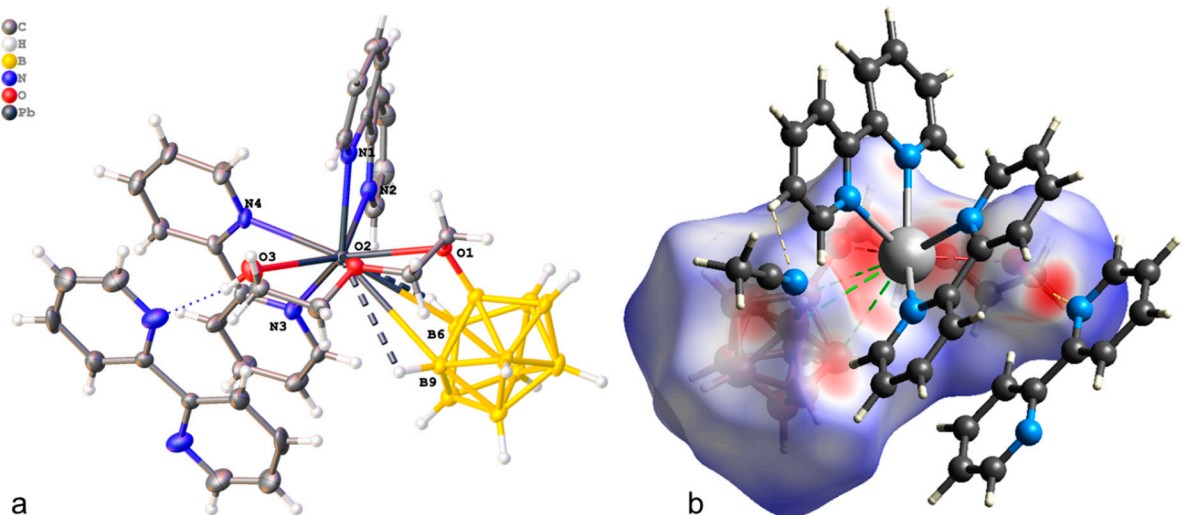

**Figure 3.** (**a**) Structure of complex **2**·0.5bipy·CH$_3$CN (**a,b**) Hirschfeld surface of the [B$_{10}$H$_9$O(CH$_2$)$_2$O(CH$_2$)$_2$OH]$^{2-}$ anion in complex **2**·0.5bipy·CH$_3$CN.

**Table 2.** Metal–heteroatom distances in complex **2**·0.5bipy·CH$_3$CN.

| Bond | Distance, Å |
|---|---|
| Pb1–O1 | 2.658(4) |
| Pb1–N1 | 2.534(4) |
| Pb1–N2 | 2.558(5) |
| Pb1–N3 | 2.725(4) |
| Pb1–N4 | 2.669(5) |
| Pb1–B6 | 2.921(5) |
| Pb1–H6A<br>Pb1–B9<br>Pb1–H9 | 3.381(6) |

In the packing of crystal **2**·0.5bipy·CH$_3$CN, the bipyridyl clathrate molecule binds two neighboring molecules of the complex via hydrogen OH . . . N bonds (O . . . N distance 2.81 Å, OH . . . N angle 167.4°) and forms two pairs of π–π stacking interactions with co-ordinated bipyridyl molecules of these complexes. The second bipyridyl molecule of complex **2**·0.5bipy·CH$_3$CN forms a π–π stacking interaction with a symmetrically equivalent molecule, resulting in the formation of a 2D polymer column parallel to the bisector of angle *ab*. The columns are interconnected by weak CH . . . HB contacts (Figure 4).

The crystallographically independent part of the monoclinic unit cell of complex **3** is built of the monomeric unit [Pb(bipy)[B$_{10}$H$_9$O(CH$_2$)$_5$O(CH$_2$)$_2$OH]] of the polymer complex [Pb(bipy)[B$_{10}$H$_9$O(CH$_2$)$_5$O(CH$_2$)$_2$OH]]$_n$ (Figure **??**a). The co-ordination polyhedron of lead includes two N atoms of a bipy ligand, an oxygen atom of the pendant hydroxy group, and two faces B1B3B4 and B6B9B10 from two adjacent boron cages. The Pb–X distances are given in Table 3. Thus, three different [B$_{10}$H$_9$O(CH$_2$)$_5$O(CH$_2$)$_2$OH]$^{2-}$ anions are co-ordinated by each metal atom, resulting in the formation of 2D polymer networks parallel to plane *bc*. On the Hirschfeld surface of the boron cluster anion (Figure **??**b), the B–Pb contacts are shown as a large red area and are indicated by dotted green lines.

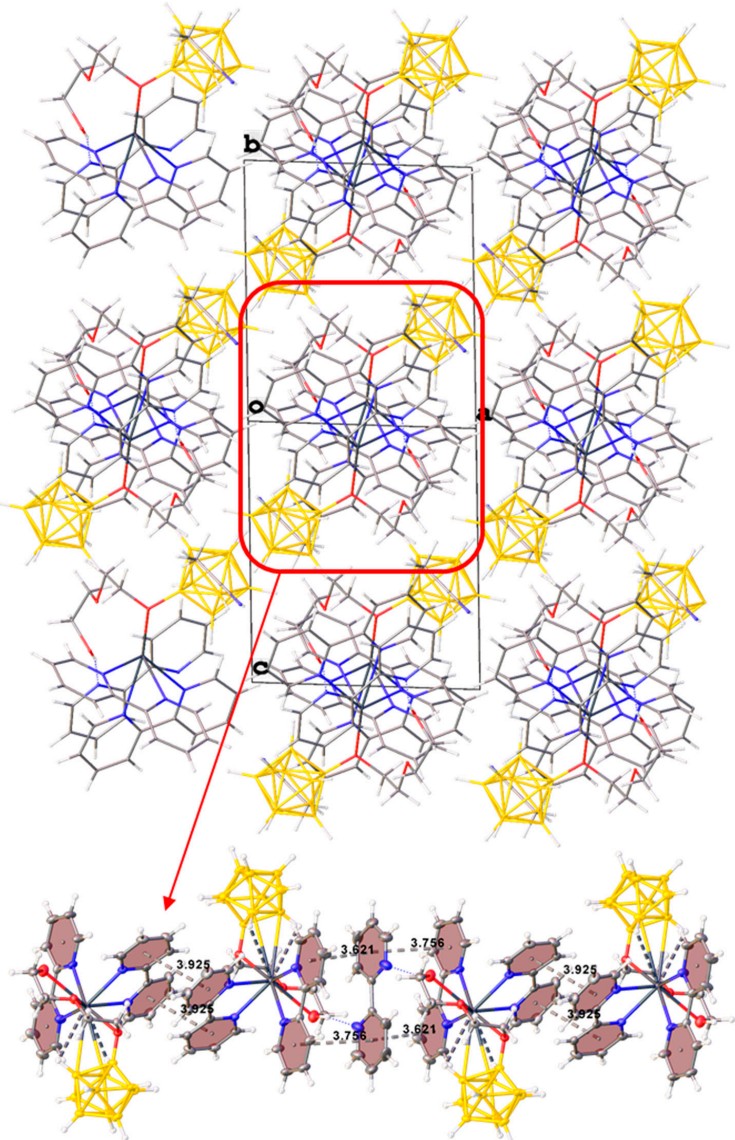

**Figure 4.** Packing diagram of **2**·0.5bipy·CH$_3$CN and $\pi$–$\pi$ stacking interactions in the crystal.

**Table 3.** Metal–heteroatom distances in complex **3**.

| Bond | Distance, Å |
|:---:|:---:|
| Pb1–O3 | 2.5109(19) |
| Pb1–N1 | 2.581(2) |
| Pb1–N2 | 2.521(2) |
| Pb1–B1 [1] | 3.053(4) |
| Pb1–B3 [1] | 3.169(4) |
| Pb1–B4 [1] | 3.003(4) |
| Pb1–B6 [2] | 2.999(3) |
| Pb1–B9 [2] | 3.215(3) |
| Pb1–B10 [2] | 3.056(3) |

[1] $x, -1 + y, z$; [2] $x, 1/2 - y, -1/2 + z$.

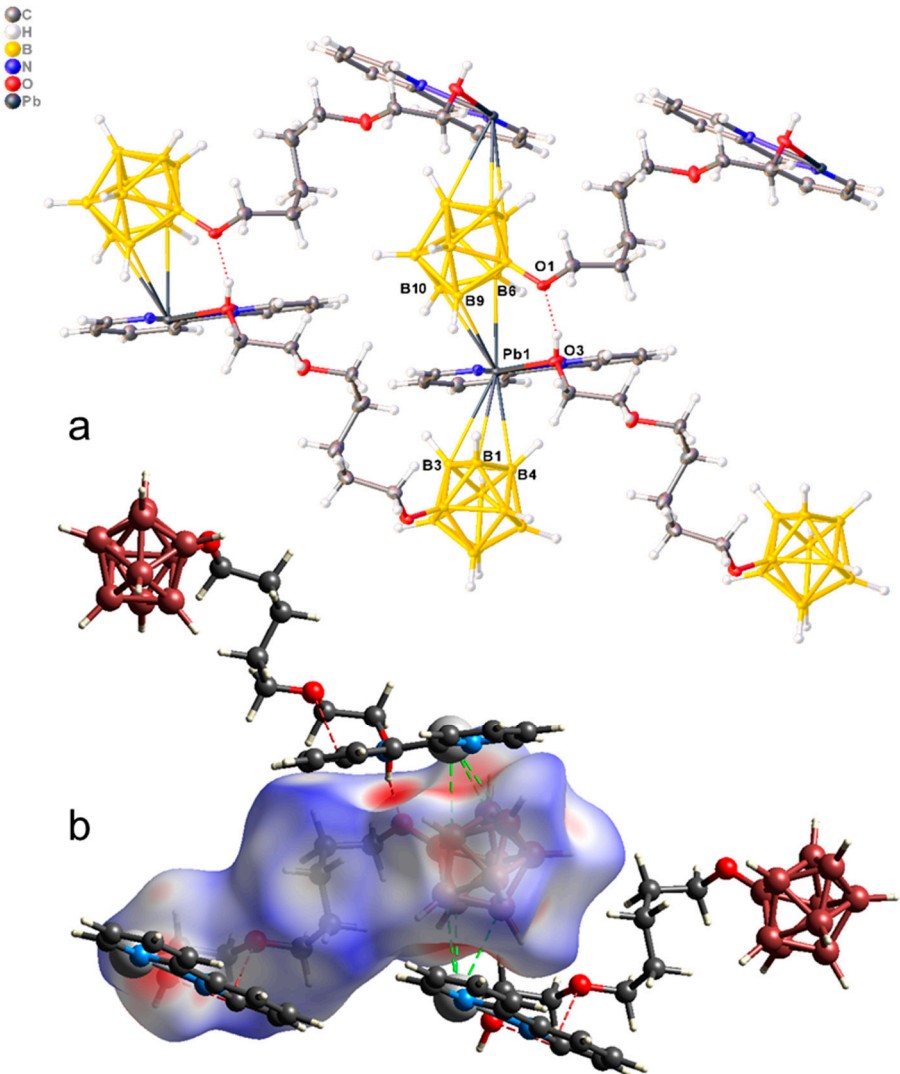

**Figure 5.** (**a**) Structure of polymeric complex **3**; (**b**) Hirschfeld surface of the $[B_{10}H_9O(CH_2)_5O(CH_2)_2OH]^{2-}$ anion in complex **3**.

Thus, three different anions $[B_{10}H_9O(CH_2)_5O(CH_2)_2OH]^{2-}$ are coordinated on each metal atom, resulting in 2D polymer networks parallel to plane *bc* (Figure 6). The anions in the planes are also additionally linked by OH . . . O hydrogen bonds (O . . . O distance 2.60 Å, angle O–H . . . O = 165.1°). The planes turned to each other by bipyridyl fragments are interconnected due to π–π stacking interactions (centroid–centroid distance 3.73 Å, angle 6.0°, shift distance 1.16 Å). Additionally, the planes are connected by dihydrogen CH . . . HB interactions between the methyl groups of the exopolyhedral substituent, the CH groups of the bipyridyl, and the BH groups of the boron cage.

It is interesting that, in spite of the fact that in all reactions we used the Pb:bipy ratio equal to 1:2, we see that in the structures of final compounds this ratio is 1:2, 1:2.5, and 1:1 for compounds **1** CH₃CN, **2** 0.5bipy·CH₃CN, and **3**, respectively. This indicates that the nature of the substituted derivatives governs the reaction path and the structure of final complexes rather than the reagents ratio.

Thus, the studied substituted derivatives of the *closo*-decaborate anion of the composition $[B_{10}H_9OH]^{2-}$, $[B_{10}H_9O(CH_2)_2O(CH_2)_2OH]^{2-}$, and $[B_{10}H_9O(CH_2)_5O(CH_2)_2OH]^{2-}$ were found to act as ligands in lead(II) complexation giving complexes of the general formula $[PbL_x[An]]$ (L = bipy; $x$ = 1, 2; [An] is the substituted derivative of the boron cluster). In all three compounds, both boron clusters and bipy ligands form the co-ordination environment of lead(II). The Pb–B bonds in crystals **1**·CH₃CN, **2**·0.5bipy·CH₃CN, and **3** are

3.317(9)–3.721(9), 2.921(5), and 2.999(3)–3.215(3) Å, respectively. Note that the data obtained agree with those found for lead(II) complexes with other O-containing substituents [56–58]. The substituted derivative $[An]^{2-}$ acts (i) as a terminal ligand in **2**·0.5bipy·$CH_3CN$ giving a mononuclear lead(II) complex, (ii) as a bringing ligand in **1**·$CH_3CN$ connecting two lead(II) atoms by both OH and BH groups giving a binuclear complex, and (iii) as a bringing ligand in **3** connecting three lead(II) atoms by BH groups and terminal OH groups giving a 2D structure. The nature of the substituted derivatives and the co-ordination capacity of the boron cage itself resulted in three various structures of the final lead(II) complexes: mononuclear, binuclear, and polymeric.

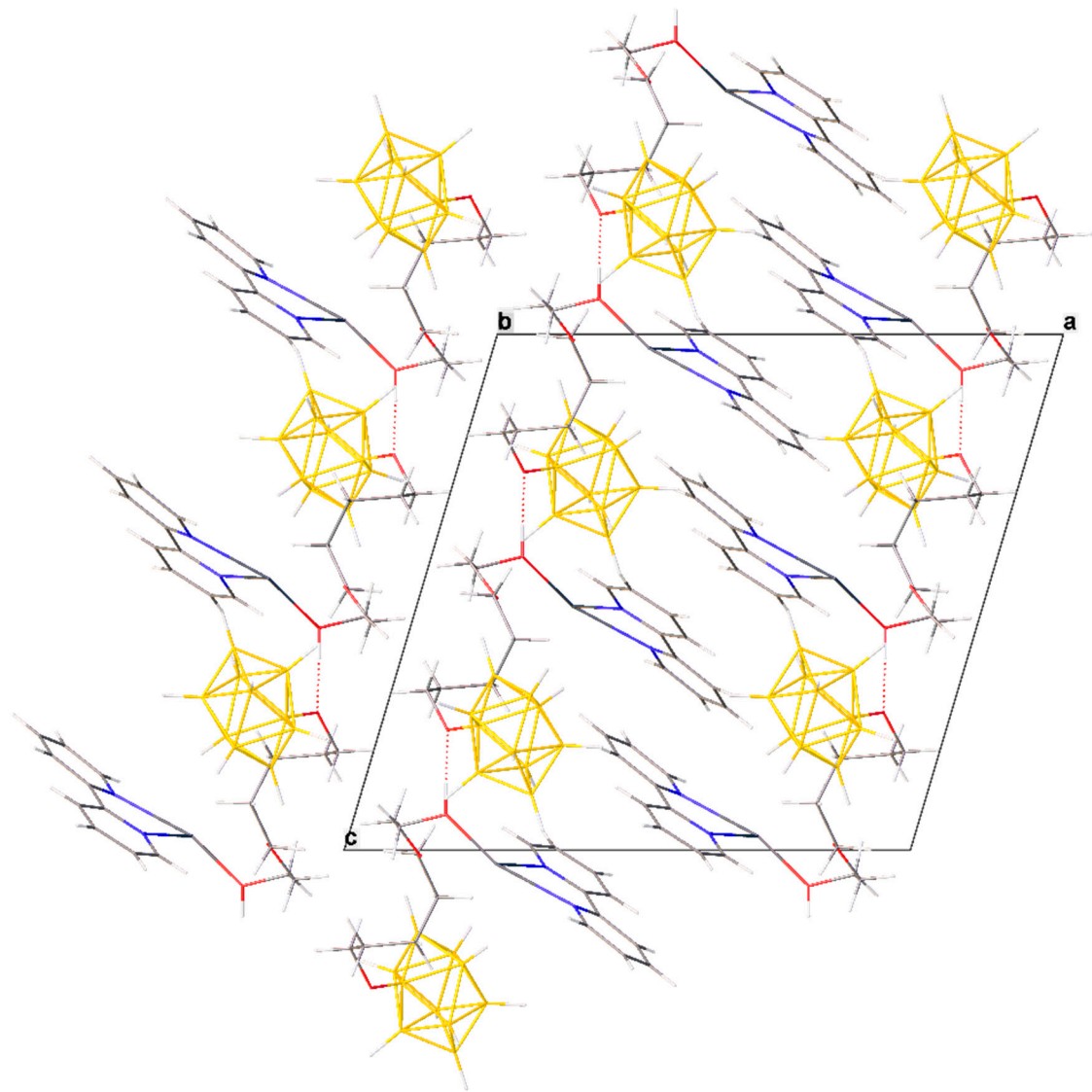

**Figure 6.** Packing diagram of **3**.

## 3. Experimental

### 3.1. Materials

Lead nitrate (99%), 2,2′-bipyridyl (bipy) (99%), aluminium chloride (anhydrous, powder, 99.99%), acetonitrile (for HPLC), tetrahydropyran (99%), dichloromethane (99%), potassium methoxide (95%), ethylene glycol (99%), ethanol (99.5%), cesium fluoride (99%), tetraphenylphosphonium chloride (99.9%) were purchased from Sigma-Aldrich (St. Louis, MO, USA) and used without additional purification. Tetraphenylphosphonium [2-hydroxy]nonahydro-*closo*-decaborate ($Ph_4P)_2[B_{10}H_9OH]$ was synthesized according to a

previously developed procedure [56]. Tetraphenylphosphonium 2-[2-(2-hydroxyethoxy)ethoxy]nonahydro-*closo*-decaborate $(Ph_4P)_2[B_{10}H_9O(CH_2)_2O(CH_2)_2OH]$ was synthesized according to the reported method [57].

### 3.2. Syntheses

#### 3.2.1. Synthesis of $K[B_{10}H_9OC_5H_{10}]$

$(Et_3NH)_2[B_{10}H_{10}]$ (0.65 g, 2.0 mmol) and $AlCl_3$ (0.53 g, 4.0 mmol) were added to tetrahydropyran (10 mL) under dry argon. The resulting suspension was boiled for 2 h, and gas evolution was observed. The reaction mixture was cooled, and the solvent was evaporated to dryness on a rotary evaporator. Dichloromethane (20 mL) was added to the obtained solution; the solution was filtered and evaporated on a rotary evaporator. The resulting white powder was dried under high vacuum at 60 °C for 1 h. Then the compound obtained and KOH (0.22 g, 4 mmol) were added to distilled water (20 mL), and the resulting solution was heated to 40 °C for 2 h. Then the resulting mixture was evaporated on a rotary evaporator, and an oil was formed. After addition of acetonitrile (10 mL) and shaking, two phases formed. The upper acetonitrile fraction was separated and the same procedure was doubly repeated. The collected solution was dried with sodium sulfate, evaporated on a rotary evaporator, and then dried in high vacuum for 2 h. Yield, 12.3 g (63%). $^1H$ NMR ($CD_3CN$, δ, ppm): 4.68 (t, 4H, $CH_2$ (α)), 2.21 (m, 4H, $CH_2$ (β)), 2.07 (m, 2H, $CH_2$ (γ)). $^{11}B$ NMR ($CD_3CN$, δ, ppm): 6.3 (s, B (2)), −0.1, −5.9 (s, B (10, 1)), −22.0, −23.8 (s, 4B, BH (3,5,6,9)), −29.9 (s, B (7,8)), −31.1 (c, B(4)). IR, $cm^{-1}$: 2470 (ν(B-H)), 943 (ν(C-O)). Found (%): C, 24.51; H, 7.84; B, 44.46; $KB_{10}C_5OH_{19}$. Calcd. (%): C, 24.77; H, 7.90; B, 44.60.

#### 3.2.2. Synthesis of $(Ph_4P)_2[B_{10}H_9OCH_2CH_2CH_2CH_2CH_2OCH_2CH_2OH]$

Tetraphenylphosphonium 2-[(2-(2-hydroxyethoxy)ethoxy)ethoxy]nonahydro-*closo*-decaborate $(Ph_4P)_2[B_{10}H_9O(CH_2)_5O(CH_2)_2OH]$ was synthesized based on the previously synthesized derivative $K[B_{10}H_9O(CH_2)_5]$ according to a procedure similar to that reported [61].

Potassium methoxide (0.28 g, 4 mmol) and ethylene glycol (10 mL) were placed under dry argon in a flask equipped with a Dean–Stark trap and heated with stirring at 100 °C until complete dissolution of $CH_3OK$. Previously prepared $K[B_{10}H_9OC_5H_{10}]$ (0.48 g, 2 mmol) was added to the resulting solution. The resulting system was heated for 1 h at a temperature of 100 °C. After cooling the reaction mixture to room temperature, a CsF solution (0.76 g, 5 mmol) in ethanol (10 mL) was added. A white precipitate was formed, which was filtered off, washed with ethanol (2 × 25 mL), and dried on a Schott filter. Next, the resulting compound was dissolved in water (20 mL), and a solution of (Ph4P)Cl (1.50 g, 4 mmol) in water (25 mL) was added. The light yellow precipitate formed was filtered off, again washed with water (2 × 25 mL), and dried in a high vacuum at 60 °C for 1 h. Yield, 1.62 g (86%).

$^1H$ NMR ($D_2O$, δ, ppm, signals from the cation are not given): 4.22 (t, 2H, -O-$CH_2$-$CH_2$-$CH_2$-$CH_2$-$CH_2$-O-$CH_2$-$CH_2$-), 4.10 (t, 2H, -O-$CH_2$-$CH_2$-$CH_2$-$CH_2$-$CH_2$-O-$CH_2$-$CH_2$-), 4.03 (t, 2H, -O-$CH_2$-$CH_2$-$CH_2$-$CH_2$-$CH_2$-O-$CH_2CH_2$-), 3.77 (t, 2H, -O-$CH_2$-$CH_2$-$CH_2$-$CH_2$-$CH_2$-O-$CH_2$-$CH_2$-), 2.06 (m, 2H, -O-$CH_2$-$CH_2$-$CH_2$-$CH_2$-$CH_2$-O-$CH_2$-$CH_2$-), 1.93 (m, 2H, -O-$CH_2$-$CH_2$-$CH_2$-$CH_2$-$CH_2$-O-$CH_2$-$CH_2$-), 1.73 (m, 2H, -O-$CH_2$-$CH_2$-$CH_2$-$CH_2$-$CH_2$-O-$CH_2$-$CH_2$-). $^{11}B\{^1H\}$ NMR (DMSO-$d^6$, δ, ppm): −1.3 (s, 1B, BO (2)), −3.1, −3.9 (both s, both 1B, BH (10, 1)), −23.7 (s, 4B, BH (3,5,6,9)), −29.2 (s, 2B, BH (7,8)), −34.4 (s, 1B, BH (4)). IR, $cm^{-1}$: 3471 (ν(OH)), 2450 (ν(BH)), 1620 (δ(OH)), 1030–1155 (ν(C–O)$_{alkox}$). Found (%): C 15.59; H 4.50; B 20.26, $C_7H_{24}B_{10}Cs_2O_3$. Calcd. (%): C 15.86; H 4.56; B 20.39. ESI MS: Found, awu.: 265.29 $\{H^+ + [B_{10}H_9O(CH_2)_5OCH_2CH_2OH]^{2-}\}^-$. $B_{10}C_7O_3H_{25}$. Calcd.: M = 265.27. Found, awu: 397.27 $\{Cs^+ + [B_{10}H_9O(CH_2)_5OCH_2CH_2OH]^{2-}\}^-$. $C_7H_{24}B_{10}CsO_3$. Calcd.: M = 397.28.

#### 3.2.3. Synthesis of Complexes **1–3**

Solid lead nitrate (0.01 mol) was added to a solution containing salt $(Ph_4P)_2[B_{10}H_9R]$ (0.01 mol) in acetonitrile (10 mL) at room temperature and vigorously stirred for 3 h. The

undissolved lead nitrate was filtered off, and a solution of bipy (0.02 mmol) in acetonitrile was added to the reaction mixture. The formation of yellow crystalline precipitates **1**–**3** was observed after 3–5 h. The crystals were filtered off and dried in air. Yield, 60–68% based on boron.

[Pb(bipy)$_2$[B$_{10}$H$_9$OH]] (**1**): NMR $^{11}$B{$^1$H} (D$_2$O, $\delta$, ppm): $-0.9$ (s, 1B, BO (2)), $-2.0$, $-3.3$ (both s, both 1B, BH (10,1)), $-21.6$ (s, 4B, BH (3,5,6,9)), $-27.0$ (s, 2B, BH (7,8)), $-32.9$ (s, 1B, BH (4)). IR (cm$^{-1}$, KBr): $\nu$(OH) 3509; $\nu$(BH) 2462br, $\nu$(BH)$_{MHB}$ ~2342sh; $\nu + \delta$(bipy) 1593, 1567, 1490, 1436, 1377, 1312, 1246, 1157, 1104, 1061, 902, 735, 642; $\pi$(CH) 766. Found (%): C, 36.47; H, 3.96; N, 8.50; B, 16.39; Pb, 31.53. C$_{20}$H$_{26}$B$_{10}$N$_4$OPb. Calcd. (%): C, 36.74; H, 4.01; N, 8.57; B, 16.54; Pb, 31.69. Crystal [Pb(Bipy)$_2$[B$_{10}$H$_9$OH]]·CH$_3$CN suitable to perform X-ray diffraction studies was taken directly from the reaction mixture.

[Pb(bipy)$_2$[B$_{10}$H$_9$O(CH$_2$)$_2$O(CH$_2$)$_2$OH]] (**2**): NMR $^{11}$B{$^1$H} (D$_2$O, $\delta$, ppm): $-1.3$ (s, 1B, BO (2)), $-2.9$, $-4.1$ (both s, both 1B, BH (10,1)), $-23.8$ (s, 4B, BH (3,5,6,9)), $-29.5$ (s, 2B, BH (7,8)), $-34.4$ (s, 1B, BH (4)). IR (cm$^{-1}$, KBr): $\nu$(OH) 3452; $\nu$(BH) 2470br, $\nu$(BH)$_{MHB}$ ~2350sh; $\nu + \delta$(bipy) 1593, 1563, 1450, 1436, 1378, 1313, 1247, 1156, 1102, 1005, 899, 641; $\pi$(CH) 763. Found (%): C, 42.57; H, 4.56; N, 8.50; B, 13.39; Pb, 25.78. C$_{29}$H$_{38}$B$_{10}$N$_5$O$_3$Pb. Calcd. (%): C, 42.48; H, 4.67; N, 8.54; B, 13.18; Pb, 25.27. Crystal [Pb(bipy)$_2$[B$_{10}$H$_9$O(CH$_2$)$_2$O(CH$_2$)$_2$OH]]·0.5bipy·CH$_3$CN suitable to perform X-ray diffraction studies was taken directly from the reaction mixture.

[Pb(bipy)[B$_{10}$H$_9$O(CH$_2$)$_5$O(CH$_2$)$_2$OH]]$_n$ (**3**): NMR $^{11}$B{$^1$H} (D$_2$O, $\delta$, ppm): $-0.1$ (s, 1B, BO (2)), $-1.8$, $-2.6$ (both s, both 1B, BH (10,1)), $-22.4$ (s, 4B, BH (3,5,6,9)), $-28.0$ (s, 2B, BH (7,8)), $-33.1$ (s, 1B, BH (4)). Found (%): C, 32.11; H, 4.99; N, 5.01; B, 17.33; Pb, 32.84. C$_{17}$H$_{32}$B$_{10}$N$_2$O$_3$Pb. Calcd (%): C, 32.53; H, 5.14; N, 4.46; B, 17.22; Pb, 33.01. IR (cm$^{-1}$, KBr): $\nu$(OH) 3456; $\nu$(BH) 2463br, $\nu$(BH)$_{MHB}$ ~2350sh; $\nu + \delta$(bipy) 1594, 1565, 1485, 1436, 1378, 1312, 1246, 1157, 1103, 1061, 902, 737, 641; $\pi$(CH) 765. Crystal [Pb(bipy)[B$_{10}$H$_9$O(CH$_2$)$_5$O(CH$_2$)$_2$OH]]$_n$ suitable for X-ray diffraction studies was taken directly from the reaction solution.

### 3.3. Methods

Elemental analysis of samples **1**–**3** for carbon, hydrogen, and nitrogen was performed on a Carlo ErbaCHNS-3 FA 1108 (Milan, Italy) automated elemental analyzer. The content of boron and metal was determined on an iCAP 6300 Duo ICP emission spectrometer with inductively coupled plasma. Samples were dried in vacuum to constant weight before the measurements; for **1**·CH$_3$CN and **2**·0.5bipy·CH$_3$CN, solvent-free samples **1** and **2**·0.5bipy were obtained.

IR spectra of compounds **1**·CH$_3$CN, **2**·0.5bipy·CH$_3$CN, and **3** were recorded on a Lumex Infralum FT-02 Fourier-transform spectrophotometer at a resolution of 1 cm$^{-1}$ in the range of 4000–600 cm$^{-1}$. Samples were prepared as suspensions in vaseline oil; NaCl pellets were used.

$^1$H and $^{11}$B NMR spectra of solutions of the obtained compounds in CD$_3$CN or D$_2$O were recorded on a Bruker DPX-300 NMR spectrometer at frequencies of 300.3 and 96.32 MHz, respectively, with internal deuterium stabilization (see Supplementary Materials, Figures S1–S7).

Mass spectra were recorded using an Agilent 1200 four-channel pump (G1311A) and a TSQ Quantum Access MAX triple quadrupole mass spectrometer equipped with an API source (HESI-II) and a 0.002 cm$^3$ 6-port external loop injector.

X-ray diffraction data for complexes **1**–**3** were collected on a Bruker D8 Venture (Centre of Shared Equipment of the Kurnakov Institute of General and Inorganic Chemistry RAS) using $\varphi$ and $\omega$-scan mode. The data obtained were indexed and integrated using the SAINT program [62]. The crystallographic data were corrected for absorption based on measurements of equivalent reflections (SADABS) [63]. The structures were determined by direct methods and refined by the full-matrix least squares technique on $F^2$ with anisotropic displacement parameters for non-H atoms. The H atoms were placed in calculated positions and refined within the riding model with fixed isotropic displacement parameters [$U_{iso}$(H) = 1.5$U_{eq}$(C) for

the $CH_3$-groups and $1.2U_{eq}(C)$ for the other groups]. All calculations were performed using the SHELXTL [64] and OLEX2 [65]. For details, see Table S1.

The crystallographic data were deposited with the Cambridge Crystallographic Data Center, CCDC nos. 2243544–2243546 for **1**–**3**, respectively. Copies of this information may be obtained free of charge from the Director, CCDC, 12 Union Road, Cambridge CHB2 1EZ, UK (Fax: +44 1223 336033; e-mail: deposit@ccdc.cam.ac.uk or www.ccdc.cam.ac.uk).

Crystal Explorer 17.5 [66] was used to perform the Hirshfeld surface analysis and analyze the interactions found within crystals of compounds **1**–**3**. The donor–acceptor groups were visualized using a standard (high) surface resolution and $d_{norm}$ surfaces are mapped over a fixed color scale of $-0.640$ (red) to $0.986$ (blue) a.u.

## 4. Conclusions

Here, three new lead(II) complexes with *closo*-decaborate anions containing monohydroxy-derivatives $[B_{10}H_9OH]^{2-}$, $[B_{10}H_9O(CH_2)_2O(CH_2)_2O(CH_2)_2OH]]^{2-}$, and $[B_{10}H_9O(CH_2)_5O(CH_2)_2OH]]^{2-}$ were prepared in the presence of bipy. Three complexes with general formula $[Pb(bipy)_x[B_{10}H_9OR]]$ ($x$ = 1, 2) were isolated and characterized by IR and multinuclear NMR spectroscopies as well as single-crystal X-ray diffraction. The substituted *closo*-decaborate anions act as (i) terminal ligand giving mononuclear complex $[Pb(bipy)_2[B_{10}H_9O(CH_2)_2O(CH_2)_2O(CH_2)_2OH]]·0.5bipy·CH_3CN$, (ii) bridging ligand connecting two lead atoms to give binuclear complex $[Pb(bipy)_2[B_{10}H_9OH]]$, and (iii) bridging ligand connecting three lead atoms affording 2D structure found in complex $[Pb(bipy)_2[B_{10}H_9O(CH_2)_5O(CH_2)_2OH]]_n$. According to X-ray diffraction data obtained here, the desired complexes show a combined co-ordination of the boron cluster via the 3c2e PbHB bonds and O atoms of the substituents; N atoms from bipy molecules complete the co-ordination sphere of the metal atom. Analysis of the Hirschfeld surfaces of the substituted anions show co-ordination of the boron cage by lead atoms as well as an extensive network of weak intra- and intermolecular non-covalent interactions which bond molecules in crystals.

**Supplementary Materials:** The following supporting information can be downloaded at: https://www.mdpi.com/article/10.3390/inorganics11040144/s1, Figure S1: $^{11}B$ {$^{1}H$} NMR spectrum of $K[B_{10}H_9OC_5H_{10}]$ in $CD_3CN$; Figure S2: $^{1}H$ NMR spectrum of $K[B_{10}H_9OC_5H_{10}]$ in $CD_3CN$; Figure S3: $^{11}B$ {$^{1}H$} NMR spectrum of $Cs_2[B_{10}H_9OCH_2CH_2CH_2CH_2CH_2OCH_2CH_2OH]$ in $D_2O$; Figure S4: $^{1}H$ NMR spectrum of $Cs_2[B_{10}H_9OCH_2CH_2CH_2CH_2CH_2OCH_2CH_2OH]$ in $D_2O$; Figure S5: $^{11}B$ {$^{1}H$} NMR spectrum of $[Pb(bipy)_2[B_{10}H_9OH]]$ in $D_2O$; Figure S6: $^{11}B$ {$^{1}H$} NMR spectrum of $[Pb(bipy)_2[B_{10}H_9O(CH_2)_2O(CH_2)_2OH]]$ in $D_2O$; Figure S7: $^{11}B$ {$^{1}H$} NMR spectrum of $[Pb(bipy)_2[B_{10}H_9O(CH_2)_5O(CH_2)_2OH]]_n$ in $D_2O$; Table S1: Crystal data and structure refinement for compounds **1**–**3**.

**Author Contributions:** E.Y.M.: Investigation, Writing—original draft preparation; V.V.A.: Methodology, Investigation, Writing—review and editing; A.S.K.: Visualization, Validation; E.A.M.: Conceptualization, Project administration; K.Y.Z.: Conceptualization; N.T.K.: Supervision. All authors have read and agreed to the published version of the manuscript.

**Funding:** This research received no external funding.

**Institutional Review Board Statement:** Not applicable.

**Informed Consent Statement:** Not applicable.

**Data Availability Statement:** No new data were created.

**Acknowledgments:** The work was performed within the framework of the State Assignment of the Kurnakov Institute of General and Inorganic Chemistry, Russian Academy of Sciences in the field of fundamental scientific research.

**Conflicts of Interest:** The authors declare that they have no known competing financial interest or personal relationships that could have appeared to influence the work reported in this paper. The authors declare that they have no conflict of interest.

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
