# Peer review of "Synthesis and Structures of Lead(II) Complexes with Hydroxy-Substituted Closo-Decaborate Anions"

_inorganics, doi:10.3390/inorganics11040144_

Round 1
Reviewer 1 Report
In the piece of work, the authors reported the syntheses and structures of the closo-decaborate-based ligands coordinated lead (II) complexes. The referee agreed to accept the work for publication after revise. The 1H and 13C NMR data needs to be provided for all the new compounds such as complexes 1-3. These results should also be discussed properly in the text. Additionally, the original spectra should be provided for supporting information.
Author Response
Dear Referee,
Thank you for your comments. Multinuclear 1H 11B and 13C would be interesting if there were no X-ray diffraction data. As we see, structures of complexes 1-3 were determined by X-ray diffraction, thus we don't need to present all the NMR studies to identify the final compounds. But starting salts are characterized by all the NMR data.
We study the salts and complexes with 1H 13C and 11B NMR spectroscopy. In 11B NMR spectra there is no changes as compared to the spectra of initial salts of the derivatives. It is obvious that complexes 1-3 dissociate in the solution and in 13C and 1H NMR spectra of 1-3 there are the same signals assigned to substituents as those seen in the spectra of starting salts. The only difference is the presence of H and C signals from BIPY molecules. Therefore, we don't see the need to present the same spectra in the SI.
Please, see the SI. Some NMR spectra are present in Figs. S1-S7.
The changes are highlighted with green color.
Reviewer 2 Report
The Paper “Synthesis and structures of lead(II) complexes with substituted derivatives of the closo-decaborate anion with pendant OH group and that attached to the boron cluster” by E. Yu. Matveeva,b, V. V. Avdeevab, *, A. S. Kubasovb, K. Yu. Zhizhina,b, E. A. Malininab and N. T. KuznetsovbHere, is well done and the compounds well characterized. However, there are grammar problems with the titles and in the text that do not help to go through the work. For instance, the title must be rewritten. It is written … as the anion with pendant OH group and that attached to the boron cluster. This does not have a meaning. I may guess, after going along the paper, what the authors intend to say but it is not right. The authors describe in the work three Pb(II) complexes whose ligands incorporate the closo-decaborate anion and an OH group. The ligands [B10H9O(CH2)2O(CH2)2O(CH2)2OH]]2– , [B10H9O(CH2)5O(CH2)2OH]]2– and [B10H9OH]2– have been utilized. With these ligands and bipy some complexes of Pb(II) have been made, isolated and characterized. Some of the complexes were characterized by single-crystal X-ray diffraction. Pb bonding through the O, and through B-H have been observed. The additional coordination sites have been occupied by the bipy ligands. Analysis of the coordination environment has been done by studying Hirschfeld surfaces.
As I indicated the work is interesting. Not too many Pb complexes incorporating closo boranes are described and much less with the crystal structures and this makes the work special. This has led me to reason what motivated the authors to study Pb(II) complexes? It is true that perovskites have opened up the field for Pb, but they bear little resemblance to the compounds proposed by the authors. Could they explain for the scientific community to see what motivated the authors to study these Pb complexes and also that they incorporate closoborates?
On the other hand they indicate that the closo species are highly stable. This certainly has to do with their 3D aromaticity as indicagted in the following papers, and they should make reference to it.
Huckel's Rule of Aromaticity Categorizes Aromatic closo Boron Hydride Clusters By:Poater, J (Poater, Jordi) [1] , [2] , [3] ; Sola, M (Sola, Miquel) [4] , [5] ; Vinas, C (Vinas, Clara) [6] ; Teixidor, F (Teixidor, Francesc) Chemistry a European Journal 2016, 22, 7437-7443 DOI10.1002/chem.201600510
3D and 2D aromatic units behave like oil and water in the case of benzocarborane derivatives
By:Poater, J (Poater, Jordi) [1] , [2] , [3] ; Vinas, C (Vinas, Clara) [4] ; Sola, M (Sola, Miquel) [5] , [6] ; Teixidor, F (Teixidor, Francesc) [4] Nature Communications 2022 Article Number3844 DOI10.1038/s41467-022-31267-7
In the application aspect, very recently has been reported the Proton Boron Fusion Reaction as an efficient therapy. The following reference should be mentioned as it has been reported in one anionic boron cluster.
Boron clusters (ferrabisdicarbollides) shaping the future as radiosensitizers for multimodal (chemo/radio/PBFR) therapy of glioblastoma By:Nuez-Martinez, M (Nuez-Martinez, Miquel) [1] ; Queralt-Martin, M (Queralt-Martin, Maria) [2] ; Munoz-Juan, A (Munoz-Juan, Amanda) [1] ; Aguilella, VM (Aguilella, Vicente M.) [2] ; Laromaine, A (Laromaine, Anna) [1] ; Teixidor, F (Teixidor, Francesc) [1] ; Vinas, C (Vinas, Clara) [1] ; Pinto, CG (Pinto, Catarina G.) [3] , [4] ; Pinheiro, T (Pinheiro, Teresa) [5] ; Guerreiro, JF (Guerreiro, Joana F.) [3] , [4] ; Mendes, F (Mendes, Filipa) [3] , [4] ; Roma-Rodrigues, C (Roma-Rodrigues, Catarina) [6] , [7] ; Baptista, PV (Baptista, Pedro, V) [6] , [7] ; Fernandes, AR (Fernandes, Alexandra R.) [6] , [7] ; Valic, S (Valic, Srecko) [8] ; Marques, F (Marques, Fernanda) [3] , [4] Journal of Materials chemistry B , 2022, 10, Page9794-9815 DOI10.1039/d2tb01818g
And application of boron clusters to the highly promising area of MOFs also should be addressed
Switchable Surface Hydrophobicity-Hydrophilicity of a Metal-Organic Framework
By:Rodriguez-Hermida, S (Rodriguez-Hermida, Sabina) [1] , [2] ; Tsang, MY (Tsang, Min Ying) [3] ; Vignatti, C (Vignatti, Claudia) [1] , [2] ; Stylianou, KC (Stylianou, Kyriakos C.) [1] , [2] ; Guillerm, V (Guillerm, Vincent) [1] , [2] ; Perez-Carvajal, J (Perez-Carvajal, Javier) [1] , [2] ; Teixidor, F (Teixidor, Francesc) [3] ; Vinas, C (Vinas, Clara) [3] ; Choquesillo-Lazarte, D (Choquesillo-Lazarte, Duane) [4] ; Verdugo-Escamilla, C (Verdugo-Escamilla, Cristobal) [4] ; Peral, I (Peral, Inmaculada) [5] ; Juanhuix, J (Juanhuix, Jordi) [6] ; Verdaguer, A (Verdaguer, Albert) [1] , [2] ; Imaz, I (Imaz, Inhar) [1] , [2] ; Maspoch, D (Maspoch, Daniel) [1] , [2] , [7] ; Planas, JG (Giner Planas, Jose) [3] Angewandte Chemie International Edition 2016, 55 Page16049-16053 DOI10.1002/anie.201609295
A complete work that deserves to be published waiting to know what motivated the authors to study the highly unusual Pb(II) complexes and to reference the new techniques where boron compounds are applied and the cause of their stability.
Author Response
Dear Referee,
Thank you for your fruitfull comments. After reviewing the enormous material concerning the coordination chemistry of boron clusters and their derivatives (ref Coord. Chem. Rev. 2022, 469, 214636) we can conclude that copper(I), silver(I) and lead(II) are the most perspective metals forming complexes with coordinated boron clusters. As we have understood, the derivatives of the boron cluster have low coordination capacity as compared to the boron clusters [BnHn]2-. In many similar reactions where B10H10 gives us complexes, the B10H9R remais unreacted.
In this study, we started with copper(I) complexation. Hovewer, some reactions with Ph3P ligand give us the starting copper(I) complex as chloride or nitrate without the B10H9R anion. Bipy in this reactions affords copper(II) that gives no CuHB bonds.
When silver(I) reaction were performed, compounds precipitate as white powders that doesn't allow us to discuss their structure. It seems that they are polymers as {Ag2[B10H9R]}n or {Ag2L2[B10H9R]}n (in the presence of ligand L). Thus, these field will be studied by us in the nearest future, but now we can not identify the compounds prepared.
Therefore, lead is the third metal from the friendly company of three commonly used Pearson's acids (Cu(I), Ag(I), Pb(II)). We succeded in isolating three compounds and discuss here their synthesis and structures.
Our motivation was added to the purpose of the article (see Introduction).
Some references was added.
The complex construction containing "OH groups...pendant...attached... " was deleted.
Reviewer 3 Report
My impression is that this paper should be submitted to a crystallographic journal as it has little actual inorganic chemistry in it.
The introduction contains 54 references and yet the Results and Discussion section only has one (self)reference in it, which is probably due to the fact that it merely lists the various interatomic distances and descriptions of the structures with little comparison to the works of other authors.
Similarly, the Introduction mentions boron neutron capture therapy with a large number of references to it, nine in all (29-37), which hardly seems relevant to this work as lead containing compounds are not likely to be of much use in medicine! In fact, sixty references in the manuscript is much too many for such a modest piece of work.
The Introduction also refers to previous work with the ten vertex closo-borate anion and [Pb(bipyr)2] cation [50-54] and says that the boron clusters are coordinated to the metal by 3c2e bonds. I find this quite unlikely, but I am unable to access the structures in the Cambridge Database to see as they have not been submitted to the CSD. The only journal article that I can access is [50] and it makes no mention of 3c2e bonds.
I am therefore unable to make a proper assessment of this paper.
Author Response
Dear Referee,
thank you for your comments. Please, see the next information
[PbL[2-B10H9O(CH2CH2)2OEt]]2·0.5DMF CCDC 282062 XAVCED
(Ph4P)[PbL[2-B10H9OC(O)CH3)2]2 CCDC 204827 LUQPIW
[PbL2[2,8-B10H8(OC(O)CH3)2) CCDC 204828 LUQPOC
[Pb(DMF)L[2-B10H9OH]]∙DMF CCDC 182269 XIFKUS (the PbHB bonds in this compound were discussed later in https://doi.org/10.1016/j.ccr.2022.214636)
The crystallographic data was added to the CCDC base with old name of the journal, Zhurnal Neorganicheskoy Khimii instead of Russ J Inorg Chem. Thus, we added CCDC nos. to the paper in order to reader to have the possibility to find the desired information.
As we are the only group who perform lead(II) complexation with closo-borate anions, thus the most close papers dealing with lead(II) complexes are those containing our early reported data.
Hovewer, we added the information were 3c2e MHB bonds were first discussed by Lipscomb [refs. 61, 62] who won The Nobel Prize in Chemistry 1976 for "his studies on the structure of boranes illuminating problems of chemical bonding". He found this type of interations when studying copper(I) complex with the [B10H10]2- anion. Now we freely operate with this type of interactions that were widely discussed for copper(I), silver(I), and lead(II) complexes (metals actig as soft acids according to the Pearson's concept). See for details the review https://doi.org/10.1016/j.ccr.2022.214636
Some references dealing with the boron neatron capture therapy were deleted.
The changes are highlighted with blue color.
Round 2
Reviewer 2 Report
The work has improved considerably in this new version, although there are some or many errors, for example already in the abstract it is indicated ...In all three c the coordination polyhedron of lead. The c I suppose indicates compounds. Something in the text needs to be improved, for example, Encouraging with the fact that a number of.... This sentence needs to be fixed. And so on. As for the references, it seems that the authors have used an automatic system because there is a great diversity of styles. In some cases two surnames, and in some cases, there are missing authors. It also seems that an automatic search for 3D aromaticity was made. For example, it is written in the manuscript ..... These aromatic 3D cages [14-18].... and I wonder what the reference 17, L. Ren, Yi Han, X. Hou, J. Wu, All are aromatic: A 3D globally aromatic cage containing five types of 2D aromatic macrocycles, Chem 2021, 7, 3442. 413 https://doi.org/10.1016/j.chempr.2021.11.003 does when there is nothing inorganic in it? Let alone dedicated to borate clusters. This reference should be deleted because it has nothing to do with the paper.
There is a problem with easy internet searches, even more so if they are not checked afterwards. On the other hand, there are now references that should have been in the first version and were not.
Therefore I suggest adapting the references to the journal's style but making them homogeneous.
Author Response
Thank you for your comments.
We have corrected the reference list to make them homogenous. Ref. 17 was deleted. Minor text editing was performed.
Reviewer 3 Report
I still think that this is more suitable to a crystallographic journal, but if the editors find it acceptable, then I do not object.
Author Response
Thank you.